# Parameter Identification Method for a Periodic Time-Varying System Using a Block-Pulse Function

Zhi Wang, Jun Wang , Jing Tian and Yu Liu *

Key Lab of Advance Measurement and Test Technology for Aviation Propulsion System,
Shenyang Aerospace University, Shenyang 110136, China
* Correspondence: liuyuvib@163.com; Tel.: +86-13998870299

**Abstract:** For periodic time-varying systems, a method of parameter identification based on the block-pulse function is presented. Firstly, the state-space equation of the system was expanded using the block-pulse function, then the recursion formula of the parameter identification of a time-varying system was obtained, according to the irrespective and orthogonal characteristics of the block-pulse function. This study provides a wide range of applications by saving time in calculation with a highly accurate method. The parameter identification was carried out by including the numerical simulation model of a three-degree freedom system and the vibration experiment results of an asymmetrical rotor system. The state space wavelet method and EMD method were compared cross-sectionally with the proposed method; this shows that the proposed method is accurate and effective, which makes it valuable in numerous applications. It also has a certain application value for several related projects.

**Keywords:** periodic time-varying system; parameter identification; block-pulse function; asymmetric rotor

## 1. Introduction

The time-varying characteristics of parameters exist in a large number of engineering structural systems, for example, the time-varying stiffness characteristics of gear transmission systems, with a change in meshing position and degree during the meshing process, along with the time-varying characteristics of the mass and stiffness of the system during the rotation of an asymmetric or cracked rotor system. The time-varying nature of the structural parameters may cause significant changes in the dynamics of the structure and may even affect the stability of system operation. Therefore, analysis of the effects of time-varying system vibration characteristics and time-varying parameters has received extensive research attention.

Traditional research on the identification of the dynamical parameters of time-invariant systems has matured [1] (pp. 119–130), while the corresponding research on time-varying systems is more difficult and remains at the forefront of research in the field of mechanics [2] (pp. 171–180). Since the 1950s, scholars have conducted systematic theoretical research on periodic knowledge-variable systems [3]; this preliminary research has mainly used differential higher-order equations to describe and solve the stability of periodic time-invariant systems. For example, P. L. Chow and K. L. Chiou [4] (pp. 315–326) proposed stability criteria for periodic solutions in nonlinear systems. With the development of control theory and electronic computers, the state space method, which is more applicable to numerical computation, has received considerable attention from scholars. The state space method started to take off in the late 1980s [5,6] (pp. 165–169, pp. 143–157) and was gradually applied to the identification of time-varying structural parameters. First, Tasker [7] (pp. 797–808) et al. verified that the state space method shows good identification speed and broad engineering application prospects by estimating the online modal parameters of 4 × 4 truss structures, then Liu [8] (pp. 149–167) extended the modal concept of linear time-varying systems and

proposed pseudo-modal parameters based on discrete time-state space models and applied them to the parameter identification of linear time-varying systems. To further speed up the state space method for the parameter identification of time-varying systems, James Durbin [9] proposed a time series analysis of state space models, then Poinot and Trige-assou [10] (pp. 2319–2333) proposed an approximate fractional integrator with recursive poles and zeros, which, in turn, derives the integer state space equations for fractional systems; however, the matrices are still large, leading to computational complexity. In an example analysis of a periodic time-varying system, Shen [11] (pp. 82–87) analyzed the response state of a tie-rod rotor system with transient periodic motion in the mid-to-high speed range, considering the inter-disk contact effects.

Scholars have dedicated a great deal of research to solving state space equations, among which the selection of the impulse function as the expansion function can greatly reduce the computational effort of solving state space equations [12]. As early as the 1960s, R.E. Kalman [13] (pp. 152–159) pointed out that the determination of linear dynamic systems can be achieved via the impulse response matrix. Leang-San Shieh [14] (pp. 383–392) first proposed to solve the state space equations via the block-pulse function, which has the properties of orthogonality and irrelevance and can reduce the computational demands of the numerical calculation of state space equations further. Murali Bosukonda [15] proposed an algebraic method for approximating the simplified state space with fractional equivalence, based on block-pulse functions, which greatly simplifies the computation in low-order linear systems; however, as the frequency in linear systems increases, the number of primary functions required for parameter identification increases and the problem of tight dynamical properties at high frequencies cannot be solved. For the use of the block-pulse function in parameter identification, Bouafoura [16] proposed a fractional-equivalent approximation to simplify the state space algebraic method, which greatly simplifies the calculation in low-order linear systems; however, with the increase in frequency in linear systems, the number of primary functions required for parameter identification increases, and the problem of tight dynamical properties at high frequencies cannot be solved. In the field of the engineering applications of block-pulse functions for parameter identification, Yaser Hosseini [17] applied the block-pulse function to the dynamic parameter monitoring of buildings, based on continuous-time state space estimation, and proposed that the numerical calculation method of block-pulse functions has the advantages of low computational cost and high accuracy. At present, in the field of control engineering and system engineering, the research applications of block-pulse functions have become increasingly mature [17–20]. In the field of rotating machinery, there are few studies about the use of the block-pulse function for parameter identification. Liao [21] (pp. 71–76) proposed a method to identify rotor unevenness, based on nonlinear support parameters. Li [22] used the mode synthesis method to analyze the geometric disorder of blade disc vibration, considering pre-stress. The use of a block-pulse function for rotating machinery in vibration parameter identification needs further research.

In this paper, we focus on the parameter identification of periodic time-varying systems in the field of mechanical vibration. First, we use the block-pulse function to expand the state-space equations of the system, then we obtain the recurrence formula for the parameter identification of time-varying systems, based on the irrelevance and orthogonality characteristics of the block-pulse function, which truncates the vibration data periodically and quickly solves the state-space equations. Subsequently, the recurrence formula is used to solve the three-degrees-of-freedom periodic time-varying system and cracked rotor experimental system, respectively.

## 2. A Parameter Identification Theory and Algorithms

Structural dynamics problems include incentives, the structure itself (including the various parameters), and response, which are called the input, systems, and output, according to the control theory argument. The parameter identification problem belongs to the first category of inverse problems of structural dynamics, which refers to finding

the various parameters describing the characteristics of a system with known inputs and outputs.

A general linear time-varying n-DOF system can be described using the following state equation:

$$\dot{x}(t) = A(t)x(t) + B(t)u(t), \ x(0) = x_0 \tag{1}$$

where $x(t) \in R^{n \times 1}$ is the state vector, $A(t) \in R^{n \times n}$ is the time-varying system matrix, $B(t) \in R^{n \times r}$ is the time-varying control matrix, and $u(t) \in R^{r \times 1}$ is the input vector. Parameter identification, in the scenario when $x(t)$ and $u(t)$ are known, is used for the parameters of $A(t)$ and $B(t)$. Based on the initial state of the system free-response data to identify $A(t)$, we set $u(t) = 0$.

*2.1. Expansion of Functions by Block-Pulse Series*

In the interval $[0, T)$, an absolute integrable function can be expanded into a set of block-pulse series [23] (pp. 569–571):

$$f(t) \approx f_1\phi_1(t) + f_2\phi_2(t) + \cdots + f_m\phi_m(t) = \sum_{i=1}^{m} f_i\phi_i(t) \tag{2}$$

where $\phi_i(t)$ is the $i$th item of the block-pulse function, which is defined as:

$$\phi_i(t) = \begin{cases} 1 & (i-1)h \leq t \leq ih, i = 1, 2, \cdots, m \\ 0 & \text{others} \end{cases} \tag{3}$$

where $h = \frac{T}{m}$ is known as the step, $m$ is the number of segments in $[0, T)$, $f_i$ is the coefficient of the $i$th item, and:

$$f_i \approx \frac{1}{h}\int_0^T f(t)\phi_i(t)dt = \frac{1}{h}\int_{(i-1)h}^{ih} f(t)dt \tag{4}$$

When $f(t)$ is smooth and $h$ is sufficiently fine, we keep the first item to approximate, then:

$$f_i \approx \frac{1}{2}[f((i-1)h) + f(ih)] \tag{5}$$

where $f((i-1)h)$ and $f(ih)$ are the function value of the $(i-1)$th and the $i$th step. We expand $x(t)$, $x_0$, $A(t)$ in Equation (1), respectively, as the block pulse function, and let $m > n$, so that:

$$x(t) = \begin{bmatrix} x_1(t) \\ x_2(t) \\ \vdots \\ x_n(t) \end{bmatrix} \approx \begin{bmatrix} x_{11} & x_{12} & \cdots & x_{1m} \\ x_{21} & x_{22} & \cdots & x_{2m} \\ \vdots & \vdots & \cdots & \vdots \\ x_{n1} & x_{n2} & \cdots & x_{nm} \end{bmatrix} \begin{bmatrix} \phi_1(t) \\ \phi_2(t) \\ \vdots \\ \phi_m(t) \end{bmatrix} \tag{6}$$

or we write it as:

$$x(t) \approx x_{\cdot 1}\phi_1(t) + x_{\cdot 2}\phi_2(t) + \cdots + x_{\cdot m}\phi_m(t) = \sum_{i=1}^{m} x_{\cdot i}\phi_i(t) \tag{7}$$

where $x_{\cdot i} = \begin{bmatrix} x_{1i} & x_{2i} & \cdots & x_{ni} \end{bmatrix}^T$.

Similarly, we write $x_0$, $A(t)$ as:

$$x_0 \approx x_0\phi_1(t) + x_0\phi_2(t) + \cdots + x_0\phi_m(t) = \sum_{i=1}^{m} x_0\phi_i(t) \tag{8}$$

$$A(t) \approx A_1\phi_1(t) + A_2\phi_2(t) \cdots + A_m\phi_m(t) = \sum_{i=1}^{m} A_i\phi_i(t) \tag{9}$$

where $\boldsymbol{x}_0 = \begin{bmatrix} x_{10} & x_{20} & \cdots & x_{n0} \end{bmatrix}^{\mathrm{T}}$, and:

$$
\begin{cases}
\boldsymbol{A}(t) = \begin{bmatrix} a_{11}(t) & a_{12}(t) & \cdots & a_{1n}(t) \\ a_{21}(t) & a_{22}(t) & \cdots & a_{2n}(t) \\ \vdots & \vdots & \cdots & \vdots \\ a_{n1}(t) & a_{n2}(t) & \cdots & a_{nn}(t) \end{bmatrix} \\[2em]
\boldsymbol{A}_i = \begin{bmatrix} a_{11}(i) & a_{12}(i) & \cdots & a_{1n}(i) \\ a_{21}(i) & a_{22}(i) & \cdots & a_{2n}(i) \\ \vdots & \vdots & \cdots & \vdots \\ a_{n1}(i) & a_{n2}(i) & \cdots & a_{nn}(i) \end{bmatrix}
\end{cases}
$$

where $i = 1, 2, \ldots, m$.

### 2.2. Identifying the Time-Varying System Matrix A(t)

When $\boldsymbol{u}(t) = 0$, we integrate the items of the left and right sides in Equation (1), to get:

$$
\boldsymbol{x}(t) - \boldsymbol{x}(0) = \int_0^t \boldsymbol{A}(\tau)\boldsymbol{x}(\tau)\mathrm{d}t \tag{10}
$$

We expand each wave pulse function into Equation (10), according to the de-correlation of the wave pulse function:

$$
\phi_i(t)\phi_j(t) = \begin{cases} \phi_i(t), & i = j \\ 0, & i \neq j \end{cases} \tag{11}
$$

Then, Equation (10) can be expressed as:

$$
\sum_{i=1}^m (\boldsymbol{x}_{\cdot i} - \boldsymbol{x}_0)\phi_i(t) = \sum_{i=1}^m \int_0^t \boldsymbol{A}_i \boldsymbol{x}_{\cdot i}\phi_i(\tau)\mathrm{d}\tau \tag{12}
$$

since:

$$
\int_0^t \phi_i(\tau)\mathrm{d}\tau \approx h \begin{bmatrix} 0 & \cdots & 0 F \underset{\underset{i}{\uparrow}}{\frac{1}{2}} & 1F \cdots F1 \end{bmatrix} \begin{bmatrix} \phi_1(t) \\ \phi_2(t) \\ \vdots \\ \phi_m(t) \end{bmatrix} \tag{13}
$$

Hence:

$$
\sum_{i=1}^m (\boldsymbol{x}_{\cdot i} - \boldsymbol{x}_0)\phi_i(t) = h\sum_{i=1}^m \left( \frac{1}{2}\boldsymbol{A}_i \boldsymbol{x}_{\cdot i} + \sum_{j=1}^{i-1} \boldsymbol{A}_j \boldsymbol{x}_{\cdot j} \right)\phi_i(t) \tag{14}
$$

When $t \in [0, T)$, the above equations are true for any value, we make sure that the corresponding coefficients are equal on both sides of the equal, and obtain:

$$
\boldsymbol{x}_{\cdot i} - \boldsymbol{x}_0 = h\left[ \frac{1}{2}\boldsymbol{A}_i \boldsymbol{x}_{\cdot i} + \sum_{j=1}^{i-1} \boldsymbol{A}_j \boldsymbol{x}_{\cdot j} \right], \quad i = 1, 2, \cdots \tag{15}
$$

when $i = 1$, $\boldsymbol{x}_{\cdot 1} - \boldsymbol{x}_0 = \frac{h}{2}\boldsymbol{A}_1 \boldsymbol{x}_{\cdot 1}$, where $\boldsymbol{x}_{\cdot 1}$ and $\boldsymbol{x}_0$ are n-dimensional column vectors, and $\boldsymbol{A}_1$ is $n \times n$ matrix. Because $n$ of independent equations cannot be solved for $n$ unknowns, we merely choose an $n$ linearly independent initial state vector:

$$
\begin{cases}
\boldsymbol{x}_0^{(1)} : \frac{h}{2}\boldsymbol{A}_1 \boldsymbol{x}_{\cdot 1}^{(1)} = \boldsymbol{x}_{\cdot 1}^{(1)} - \boldsymbol{x}_0^{(1)} \\
\boldsymbol{x}_0^{(2)} : \frac{h}{2}\boldsymbol{A}_1 \boldsymbol{x}_{\cdot 1}^{(2)} = \boldsymbol{x}_{\cdot 1}^{(2)} - \boldsymbol{x}_0^{(2)} \\
\qquad\qquad \vdots \\
\boldsymbol{x}_0^{(n)} : \frac{h}{2}\boldsymbol{A}_1 \boldsymbol{x}_{\cdot 1}^{(n)} = \boldsymbol{x}_{\cdot 1}^{(n)} - \boldsymbol{x}_0^{(n)}
\end{cases} \tag{16}
$$

Therefore:

$$\frac{h}{2}A_1X_1 = X_1 - X_0 \tag{17}$$

where:

$$X_{.1} = \begin{bmatrix} x_{.1}^{(1)} & x_{.1}^{(2)} & \cdots & x_{.1}^{(n)} \end{bmatrix}, X_0 = \begin{bmatrix} x_{.1}^{(1)} & \cdots & x_0^{(n)} \end{bmatrix}$$

For each of the different initial state vectors, the m subintervals status response can be calculated. Since $X_0$ is an $n \times n$ full-rank matrix, $X_{.i}$ can be proved with an $n \times n$ full-rank matrix. Using Equation (17), $A_1$ can be solved:

$$A_1 = \frac{2}{h}(X_{.1} - X_0)X_{.1}^{-1} \tag{18}$$

since:

$$\frac{h}{2}A_1X_{.i} = (X_{.i} - X_0) - h\sum_{j=1}^{i-1} A_jX_{.j} \tag{19}$$

The recursive formula of system matrix $A_i$ can be obtained, as follows:

$$\begin{cases} A_1 = \frac{2}{h}(X_{.1} - X_0)X_{.1}^{-1} \\ A_{i+1} = \left[\frac{2}{h}\left(X_{.(i+1)} - X_{.i}\right) - A_iX_{.i}\right]X_{.(i+1)}^{-1} \end{cases} \tag{20}$$

*2.3. Identification of the Stiffness and Damping Matrix of the Periodic Time-Varying Vibration System $A(t)$*

A dynamic differential equation of $n$-degrees of freedom with a damping and stiffness cycle time-varying vibration system can be shown as follows:

$$M\ddot{z} + C(t)\dot{z} + K(t)z = f \tag{21}$$

where $z$ is the displacement column vector of the order $p \times 1$, and $M, C$, and $K$ are the mass, damping, and stiffness matrices of the order $p \times p$. In Equation (21), $C(t) = C_c + C_v(t)$, $K(t) = K_c + K_v(t)$, and $K_c, K_v$ are the constant and time-varying parts, respectively, and $K_v(t) = K_v\left(t + \widetilde{T}\right)$, $\widetilde{T}$ is the time-varying period in a damping analogy.

For a free vibration system, we describe Equation (21) in a state equation to be:

$$\dot{\vec{z}}(t) = A(t)\vec{z}(t), \vec{z}(0) = \vec{z}_0, \vec{z} = \begin{bmatrix} z & \dot{z} \end{bmatrix}^{\mathrm{T}} \tag{22}$$

$$A(t) = \begin{bmatrix} 0 & I \\ -M^{-1}K(t) & -M^{-1}C(t) \end{bmatrix} \tag{23}$$

where $A \in \mathrm{R}^{n \times n}, \widetilde{z} \in \mathrm{R}^{n \times 1}$, and $n = 2p$.

A square matrix is formed by $n$ linearly independent state vectors $\vec{z}_i$ ($i$ time):

$$Z_{.i} = \begin{bmatrix} \vec{z}_{.i}^{(1)} & \vec{z}_{.i}^{(2)} & \cdots & \vec{z}_{.i}^{(n)} \end{bmatrix} \tag{24}$$

It can be seen from Equation (23) that the last $p$ lines of $A$ are the parameter to be calculated. To reduce the amount of calculation and improve the speed, the matrix can be divided into blocks:

$$A_i = \begin{bmatrix} A_{11,i} & A_{12,i} \\ A_{21,i} & A_{22,i} \end{bmatrix}, Z_{.i} = \begin{bmatrix} Z_{11,.i} & Z_{12,.i} \\ Z_{21,.i} & Z_{22,.i} \end{bmatrix}$$

$$Z_{.i}^{-1} = \begin{bmatrix} \hat{Z}_{11,.i} & \hat{Z}_{12,.i} \\ \hat{Z}_{21,.i} & \hat{Z}_{22,.i} \end{bmatrix}, A_{jk,i}, Z_{jk,.i}, \hat{Z}_{jk,.i} \in \mathrm{R}^{p \times p}$$

According to Equation (20), the damping and stiffness parameter identification recursive Equations (25)–(28) can be obtained, as follows:

$$K_1 = -\frac{2}{h}M\left[(Z_{21,\cdot1} - Z_{21,0})\hat{Z}_{11,\cdot1} + (Z_{22,\cdot1} - Z_{22,0})\hat{Z}_{21,\cdot1}\right] \tag{25}$$

$$C_1 = -\frac{2}{h}M\left[(Z_{21,\cdot1} - Z_{21,0})\hat{Z}_{12,\cdot1} + (Z_{22,\cdot1} - Z_{22,0})\hat{Z}_{22,\cdot1}\right] \tag{26}$$

$$
\begin{aligned}
K_{i+1} = &\left[-\tfrac{2}{h}M\left(Z_{21,\cdot(i+1)} - Z_{21,\cdot i}\right) - (K_iZ_{11,\cdot i} + C_iZ_{21,\cdot i})\right]\hat{Z}_{11,\cdot(i+1)} \\
&+ \left[-\tfrac{2}{h}M\left(Z_{22,\cdot(i+1)} - Z_{22,\cdot i}\right) - (K_iZ_{12,\cdot i} + C_iZ_{22,\cdot i})\right]\hat{Z}_{21,\cdot(i+1)}
\end{aligned} \tag{27}
$$

$$
\begin{aligned}
C_{i+1} = &\left[-\tfrac{2}{h}M\left(Z_{21,\cdot(i+1)} - Z_{21,\cdot i}\right) - (K_iZ_{11,\cdot i} + C_iZ_{21,\cdot i})\right]\hat{Z}_{12,\cdot(i+1)} \\
&+ \left[-\tfrac{2}{h}M\left(Z_{22,\cdot(i+1)} - Z_{22,\cdot i}\right) - (K_iZ_{12,\cdot i} + C_iZ_{22,\cdot i})\right]\hat{Z}_{22,\cdot(i+1)}
\end{aligned} \tag{28}
$$

According to the theory of Sylvester and Fourier, the spectrum of the free response of a parametrically excited system is established by Equation (29):

$$f_{d,i}^c = \left|\overline{f}_d + i \cdot \widetilde{f}\right|, \ i = 0, \pm 1, \pm 2, \cdots \tag{29}$$

where $f_{d,i}^c$ and $\overline{f}_d$ are the freedom frequency response and the natural frequency of a parametric system, respectively, and $\widetilde{f}$ represents the parametric excitation frequency [24,25]. It is easy to see that a parametrically excited system has multi-frequency characteristics. When $i$ is larger, an $f_i^c$ value corresponding to the free vibration of the system is very small. When frequency components are appropriately cut off, the free vibration response of the system is still approximately cyclical, so few cycles can be used to identify the parameters. There is, then, no need to perform the whole calculation, thus greatly reducing the amount of computation necessary.

## 3. Parameter Identification of Multi-Degree of a Freedom Simulation System

As shown in Figure 1, a three-degree-of-freedom spring-mass-damping time-varying structure with mass blocks, where the mass of blocks ($m_1$ = 1 kg, $m_2$ = 2 kg and $m_3$ = 1 kg) do not vary with time. Stiffness can be shown as $k_1 = 2 \times 10^5$ N/m, $k_2 = 2 \times 10^5$ N/m, $k_c = 1 \times 10^5$ N/m, and $k_v = 2 \times 10^4$ N/m, stiffness varying frequency, $\Omega = 25.13$ rad/s, and initial phase $\varphi = \pi/2$. The time-varying stiffness is $k_3 = k_c + k_v \cos(\Omega t + \varphi)$. To highlight the study of stiffness variation, damping is assumed to be $c_1 = c_2 = c_3 = 0$ and the external excitation force is assumed to be $f_1 = f_2 = f_3 = 0$.

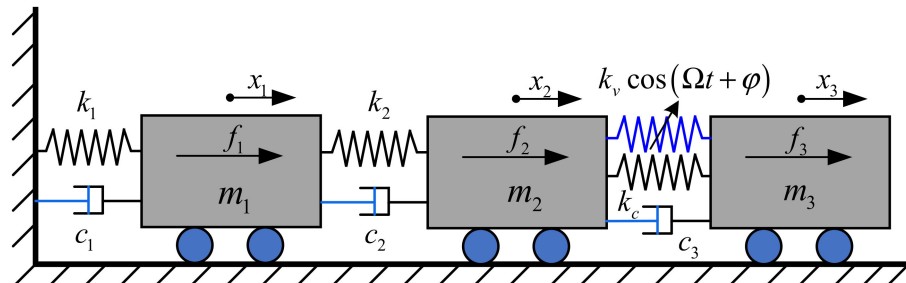

**Figure 1.** Three-degree-of-freedom system with stiffness cycle changing.

We assume that the initial situation is $x_1(0)$ = 0.001 m, $\dot{x}_1(0)$ = 0 m/s, $x_2(0)$ = 0 m/s, $\dot{x}_2(0)$ = 0 m/s, $x_3(0)$ = 0.005 m, and $\dot{x}_3(0)$ = 0 m/s. The dynamic response of the system is solved using the Newmark-$\beta$ method, taking the step size as $h = 10^{-4}s$; the displacement response is shown in Figure 2 and the velocity response is shown in Figure 3.

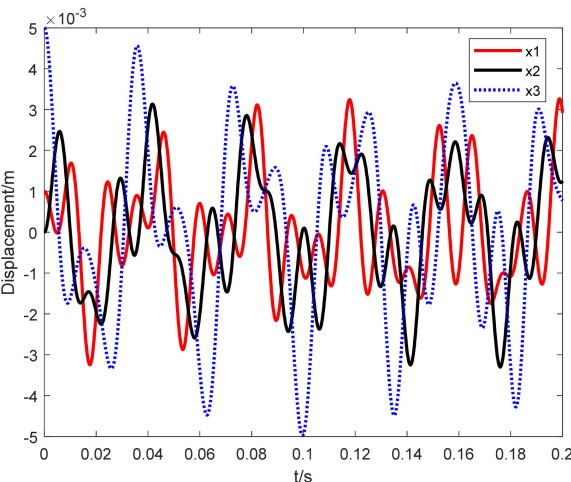

**Figure 2.** Vibration displacement response of a three-degree-of-freedom stiffness periodic time-varying system.

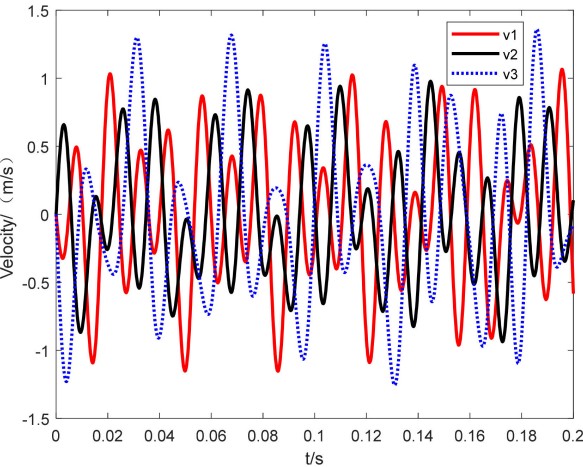

**Figure 3.** Vibration velocity response of a three-degree-of-freedom stiffness periodic time-varying system.

Spectral analysis of the displacement signal is shown in Figure 4.

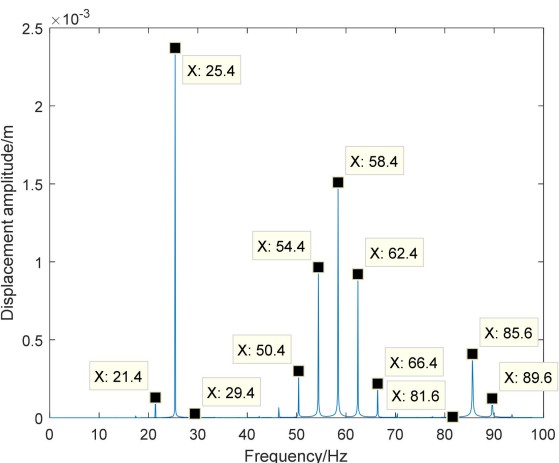

**Figure 4.** A three-degree-of-freedom stiffness periodically time-varying system vibration displacement spectrum.

It can be seen that the free response of the vibration system has such frequency components as $25.4 \pm 4.0$ Hz, $58.4 \pm 4.0$ Hz, $58.4 \pm 2 \times 4.0$ Hz, $85.6 \pm 4.0$ Hz, and so on, in accordance with Equation (29), where $\overline{f}_1 = 25.4$ Hz is the first-order intrinsic frequency, $\overline{f}_2 = 58.4$ Hz is the second-order intrinsic frequency, $\overline{f}_3 = 85.6$ Hz is the third-order intrinsic frequency, and $\widetilde{f} = \Omega/(2\pi) = 4.0$ Hz is the parametric excitation frequency. It has been verified that the parametric vibration system has multi-frequency characteristics.

In this paper, a total of six sets of response calculations under different initial states (and linearly independent) are carried out; the parameter identification program is prepared using the MATLAB language, according to the recursive Equations (25)–(28), and the results and relative errors of each stiffness identification are shown as $k_2^*$ ('$*$' refers to the identification stiffness) in Figures 5–8.

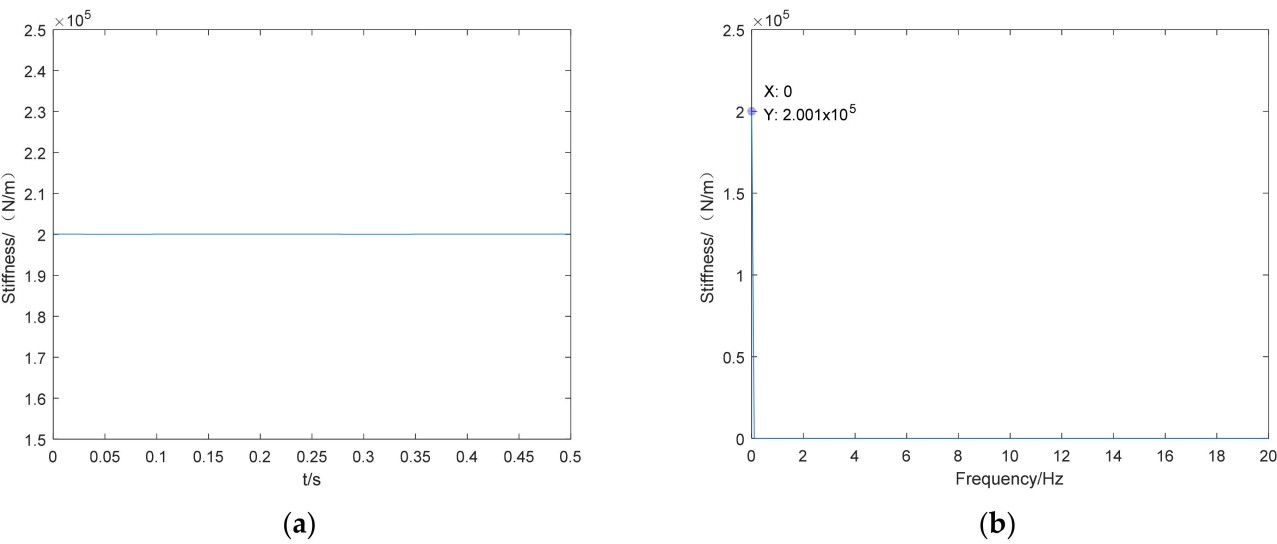

**Figure 5.** Time domain identification result (**a**) and frequency domain identification results (**b**) of non-time-varying stiffness $k_2^*$.

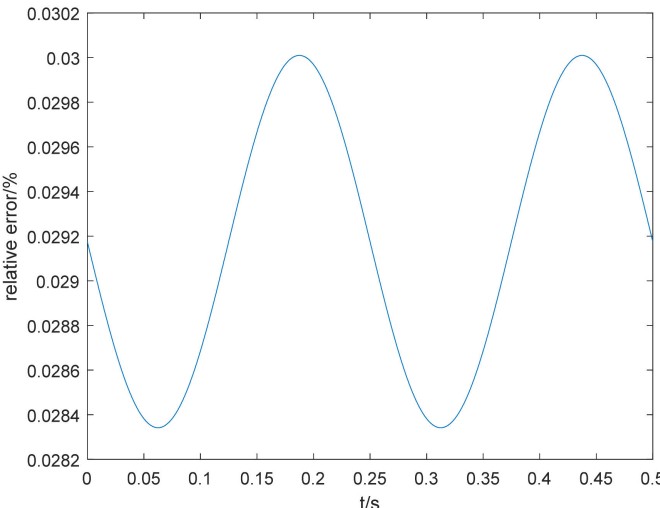

**Figure 6.** Relative error of the non-time-varying stiffness $k_2^*$ identification results.

From Figures 5 and 6, it can be seen that the parameter identification method proposed in this paper can better identify the non-time-varying stiffness value of the system, and the error is small compared with the preset value.

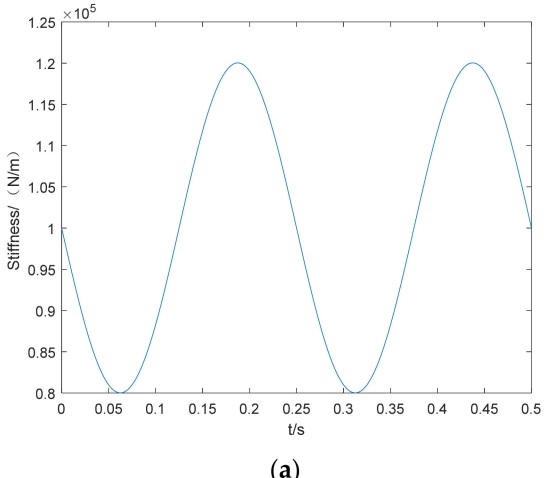 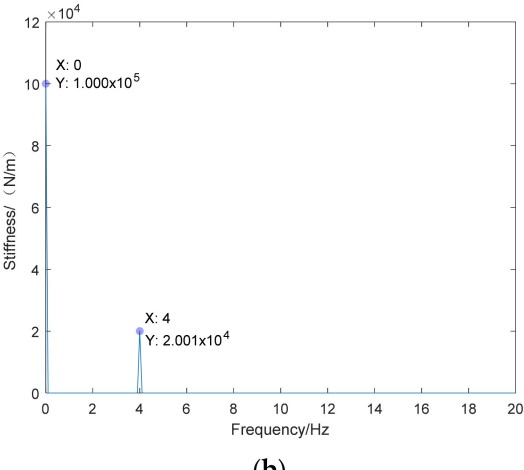

(**a**)                                    (**b**)

**Figure 7.** Time domain identification result (**a**) and frequency domain identification results (**b**) of time-varying stiffness $k_3^*$.

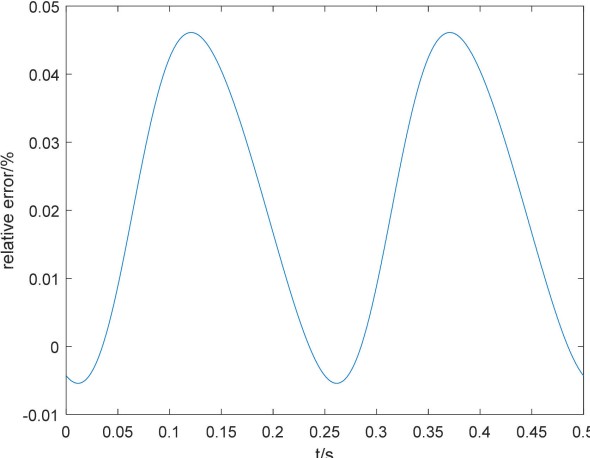

**Figure 8.** Relative error of the time-varying stiffness $k_3^*$ identification results.

From Figures 7 and 8, it can be seen that the parameter identification method proposed in this paper also identifies the time-varying stiffness value of $k_3^*$ well. It can also be seen from the right-hand panel of Figure 7 that the stiffness $k_3^*$ consists of two parts: one is the constant value (zero frequency) $1 \times 10^5$ N/m, and the other is the frequency $f = 4.0$ Hz, while the amplitude is $2.001 \times 10^4$ and its angular frequency $\widetilde{\Omega} = 2\pi f = 25.13$ rad/s is consistent with the given $\Omega = 25.13$ rad/s. As can be seen from Figure 8, the relative error in the stiffness identification results is small, with a maximum value of less than 0.05%.

To study the influence of different signal-to-noise ratios and calculation steps on the accuracy of recognition results, the mean absolute percentage error (MAPE) of recognition results is defined as:

$$MAPE = \frac{1}{N} \sum_{i=1}^{N} \left| \frac{\widetilde{p}_i - p_i}{p_i} \right| \times 100\%$$

where $\widetilde{p}_i$ and $p_i$ are the stiffness identification value and the theoretical value at moment $i \cdot h$, respectively. $N$ presents the number of samples.

We then add noise interference to the simulation signal. In the step size of $h = 10^{-4}$ s, the simulation results and relative errors of time-varying stiffness $\widetilde{p}_3$ under different signal-to-noise ratios are shown in Figures 9–11.

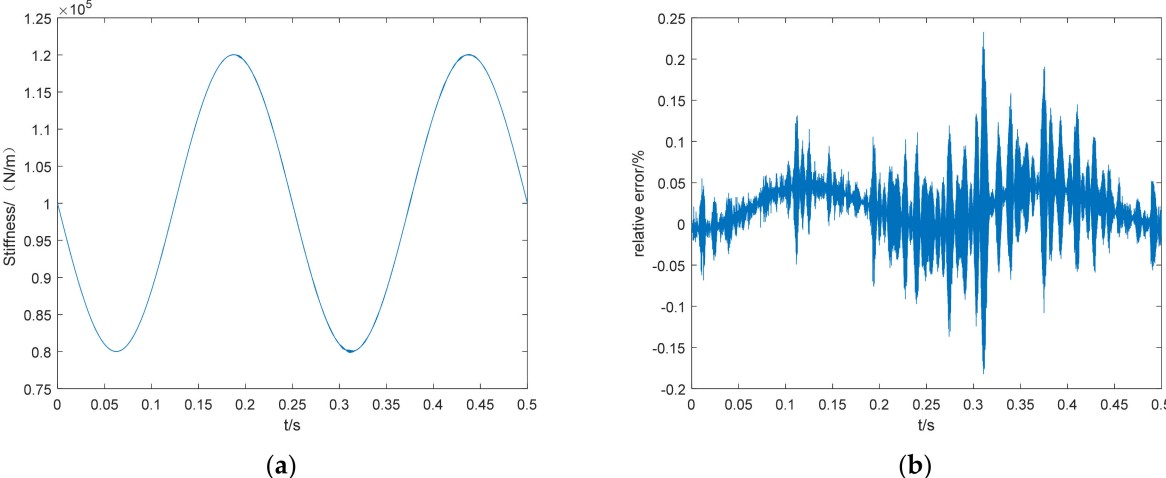

**Figure 9.** Time domain identification result (**a**) and relative error (**b**) of time-varying stiffness $\widetilde{p}_3$ at 100 dB signal-to-noise ratio.

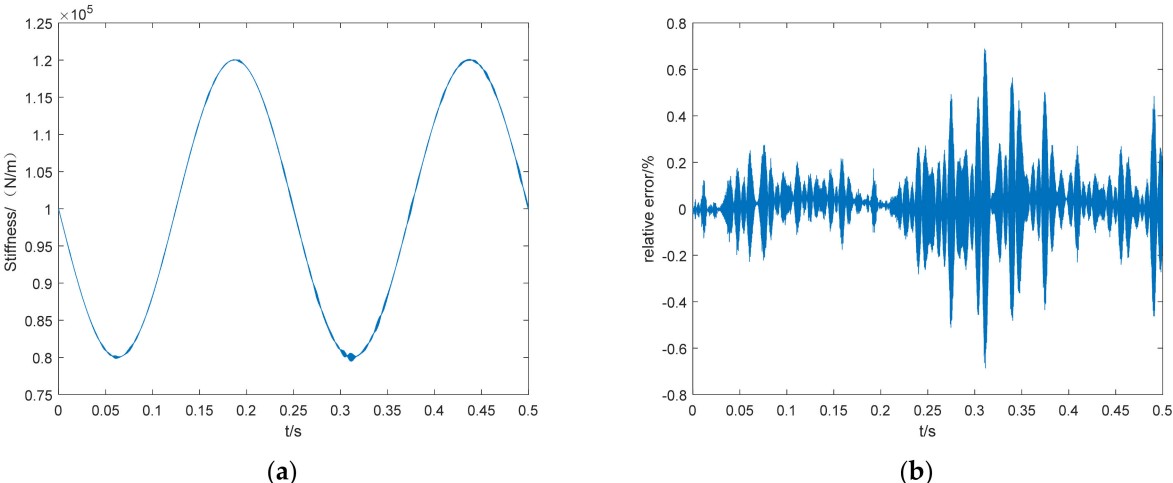

**Figure 10.** Time domain identification result (**a**) and relative error (**b**) of time-varying stiffness $\widetilde{p}_3$ at 80 dB signal-to-noise ratio.

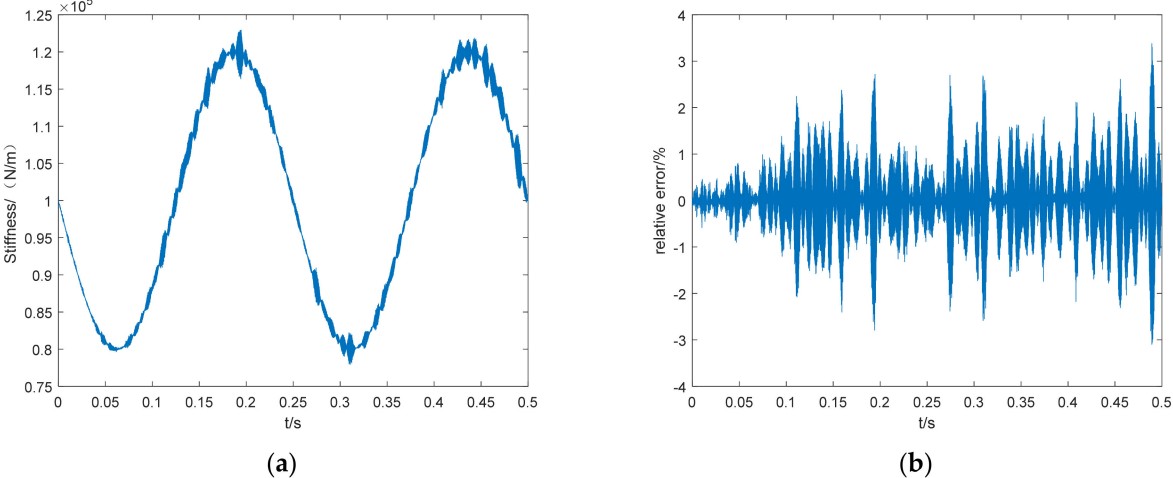

**Figure 11.** Time domain identification result (**a**) and relative error (**b**) of time-varying stiffness $\widetilde{p}_3$ at 50 dB signal-to-noise ratio.

The effects of signal-to-noise ratio and calculation step size on the recognition accuracy (MAPE) are shown in Tables 1 and 2, respectively.

**Table 1.** Stiffness identification MAPE at different SNRs.

| Signal-to-Noise Ratio (SNR) (dB) | Mean Absolute Percentage Error (MAPE) (%) |
|:---:|:---:|
| No noise | 0.0223 |
| 100 | 0.0878 |
| 80 | 0.1849 |
| 50 | 0.9703 |

**Table 2.** Stiffness identification MAPE at different step sizes without noise.

| Calculation Step (s) | Mean Absolute Percentage Error (MAPE) (%) |
|:---:|:---:|
| $10^{-3}$ | 2.1385 |
| $10^{-4}$ | 0.0223 |
| $10^{-5}$ | 0.0016 |

From Tables 1 and 2, it can be seen that the signal noise has a greater impact on the recognition result, so the signal-to-noise ratio should be improved as much as possible in the signal acquisition process; the smaller the calculation step, the better the recognition effect. However, the corresponding calculation volume also increases, so the appropriate step length should be selected according to the actual requirements.

For the Newmark-$\beta$ method used in solving the dynamic response of the system, the calculation step size affects the accuracy of the calculation, as can be seen in Figure 4 in line 186. The highest order intrinsic frequency of the three-degree-of-freedom system in this paper is $f_3 = 85.6$ Hz; according to the sampling theorem, the sampling frequency should be greater than the signal analysis frequency by more than two times so as not to occur in the mixing. In order to maintain the accuracy of the signal amplitude, sampling frequency should be much greater than $f_3$. The sampling frequency selected in this paper is $f_s = 10,000$ Hz, that is, $h = 10^{-4}$ s. At the same time, according to the results of Table 2, in line 230, it can be seen that when $h = 10^{-3}$ s, the accuracy is poor; when we take $h = 10^{-4}$ s, the accuracy has met the requirements, while if we take $h = 10^{-5}$ s, the computational volume is large. Taking this into account, we chose $h = 10^{-4}$ s.

In order to verify the higher accuracy of the proposed method, compared with the existing methods, a cross-sectional comparison of the identification results of the periodic time-varying stiffness $k_1$ is presented in the literature [26]. The paper presents and discusses the identification accuracy of the state space wavelet method for the parametric time-varying stiffness in the two-degrees-of-freedom simulation example, and shows the identification accuracy of the periodic time-varying stiffness at four signal-to-noise ratios: no noise, 50, 80, and 100, where the variation pattern of the periodic time-varying stiffness $k_1$ is the same as that of the time-varying stiffness $k_3^*$ recognized in this paper. The difference lies in the fact that the simulation example given in the literature has one less degree of freedom than this paper.

In contrast, in the case of noiseless and medium-high signal-to-noise ratios, using the block-pulse function method to identify periodic time-varying stiffness can greatly improve the identification accuracy, and the higher the signal-to-noise ratio, as shown in Table 3, the higher the accuracy of the identification results; in the absence of noise, the block-pulse function method can also identify the time-varying parameters more accurately.

In order to test the improvement of the parameter identification speed of the block-pulse function method, the parameter identification speed of this method is compared with that of the state space wavelet method and that of the EMD method; the time required to identify the time-varying stiffness sequence in a three-degree-of-freedom system using this method is recorded, while the time required to identify the time-varying stiffness sequence in the same system without noise is recorded with the same arithmetic power.

**Table 3.** Stiffness identification MAPE in two methods at different SNRs.

| Signal-to-Noise Ratio (SNR) (dB) | MAPE in Block-Pulse Function Method (%) | MAPE in State Space Wavelet Method (%) |
| :---: | :---: | :---: |
| No noise | 0.0223 | 0.0991 |
| 100 | 0.0878 | 0.3260 |
| 80 | 0.1849 | 0.4105 |
| 50 | 0.9703 | 0.6995 |

By identifying the time-varying stiffness of the same three-degree-of-freedom system, as shown in Table 4, the time required to identify the time-varying stiffness sequence by this method is 248 ms, while the time required by the state-space wavelet method is 532 ms, the time required by the EMD method is 668 ms; the speed of identification by this method is improved by 53.38% compared to the state-space wavelet method, and the speed of identification by the EMD method is improved by 62.87%.

**Table 4.** Calculating time in the three methods used in the same three-degree-of-freedom system.

| Method | Block-Pulse Function Method | State Space Wavelet Method | EMD Method |
| :---: | :---: | :---: | :---: |
| Calculate time | 248 ms | 532 ms | 668 ms |

## 4. Periodic Time-Varying Rotor System Stiffness Identification

### 4.1. Typical Asymmetric (Square Axis) Rotor Experimental System Construction

In order to experimentally verify the parameter identification method proposed in this paper, a typical asymmetric (square-axis) rotor experimental system is built, as shown in Figure 12. The instruments and equipment used in the experiment are shown in Table 5. After hammering the shaft center by 45 degrees with a horizontal axis, position the next disc. LMS Test.lab is used for data acquisition and processing.

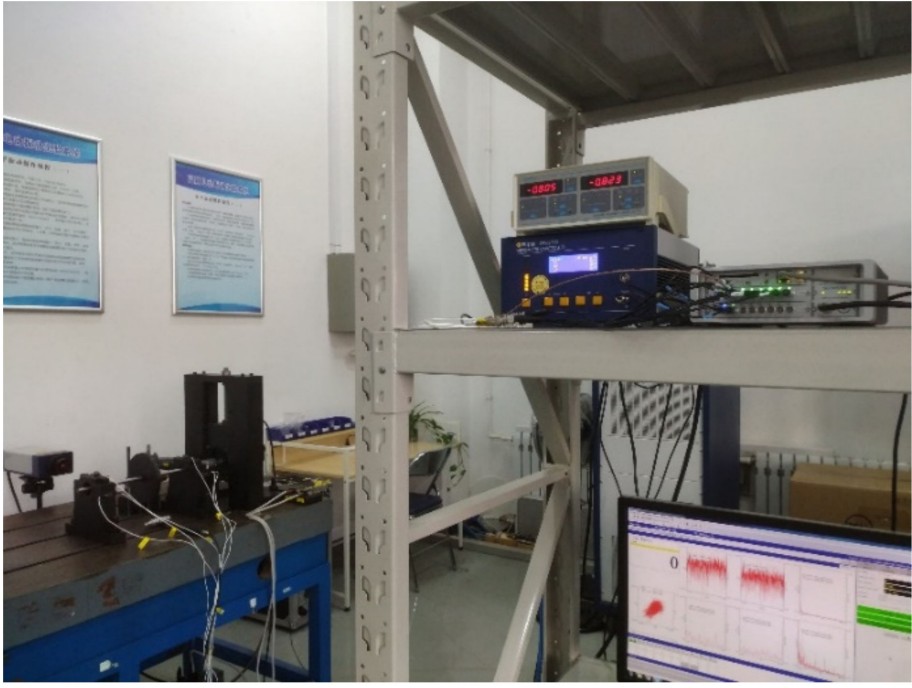

**Figure 12.** Experimental system of an asymmetric rotor.

**Table 5.** Experimental equipment.

| Instrument Name | Instrument Model |
|---|---|
| Multi-channel dynamic testing system | LMS SCADAS Recorder (SCR202) |
| Servo motor | 130SM04030 |
| Servo motor controller | JYT-SERVO-II |
| Workstation (Dell laptop) | Dell M5810 |
| Eddy current displacement measuring instrument | YE5937B |
| Eddy current displacement sensor | CWY-DO-502 |
| Laser speed tester | Polytec OFV-5000 |
| PCB acceleration sensor | 333B30 |
| PCB hammer | 086D05 |

*4.2. Square Shaft Rotor Stiffness Identification*

The structure of the square shaft rotor is shown in Figures 13 and 14, the material of the rotor shaft is 30CrMnSi; the material of the disc is 45# steel, and the relevant parameters are: $l = 300$ mm, $d_{shaft} = 14$ mm, $h = 10$ mm, $\rho_{shaft} = 7750$ kg/m$^3$, $E_{shaft} = 2.01 \times 10^{11}$ Pa, $\mu_{shaft} = 0.227$, $D_{disc} = 160$ mm, $\rho_{disc} = 7890$ kg/m$^3$, $\mu_{disc} = 0.269$. In order to achieve asymmetry in the shaft, a partial cut was made on the shaft so that the bending stiffness in the two mutually perpendicular directions ($\eta$ shaft to $\xi$ shaft) was different. In the experiments, the speed controller is adjusted to make the rotor run steadily at $n = 720$ r/min, so that the flexural stiffness of the shaft in both horizontal and vertical directions then varies with the speed; the impact response experiments are performed on the rotor system with a force hammer, to eliminate the effect of an unbalanced response of the rotor under centrifugal force by reducing the rotor unbalance and filtering, and to highlight the free response of the rotor after impulse excitation.

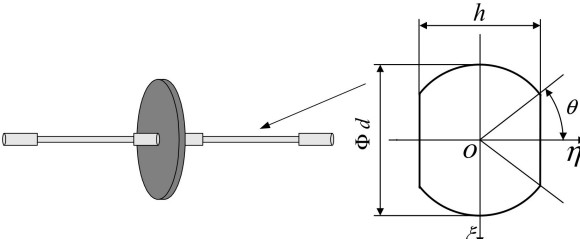

**Figure 13.** Schematic diagram of the asymmetric rotor.

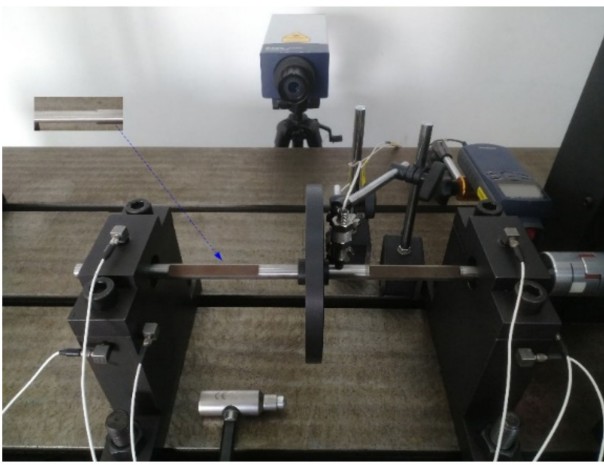

**Figure 14.** Experimental system of the asymmetric rotor.

The square shaft rotor system was hammered for several tests, and the vibration displacement at the shaft center (in both horizontal and vertical directions) was recorded

using eddy current displacement sensors; the vibration velocity at the shaft center in the horizontal direction was collected using laser velocity testers, and the vibration acceleration at the rotor support (horizontal, vertical, and axial directions) was recorded using acceleration sensors. The horizontal displacement and velocity response of a certain test are shown in Figures 15 and 16, respectively. The horizontal stiffness variation of the square shaft rotor is shown in Figure 17.

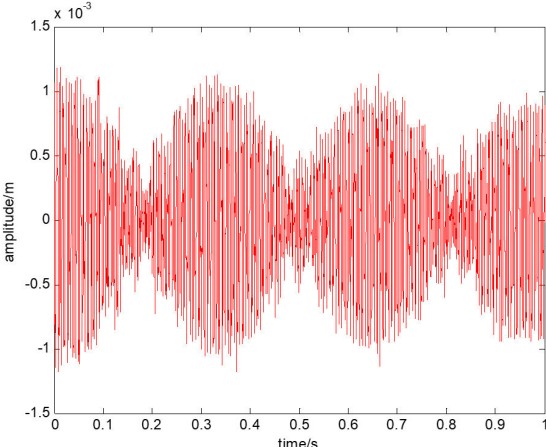

**Figure 15.** Horizontal displacement response.

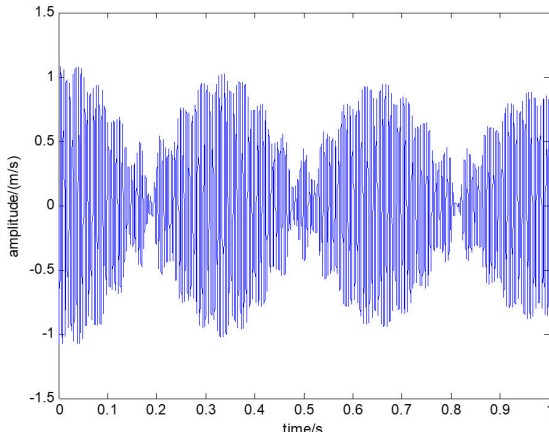

**Figure 16.** Horizontal speed response.

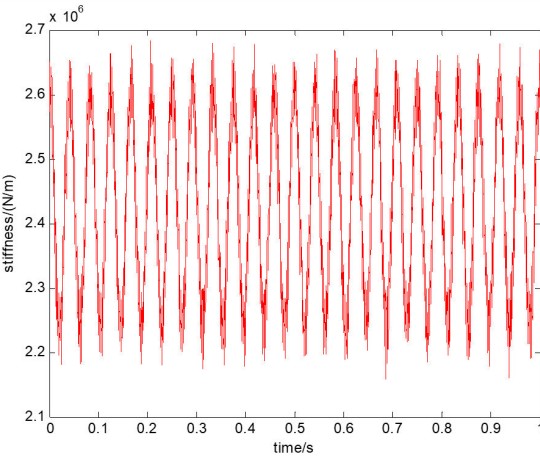

**Figure 17.** Time domain diagram of the horizontal stiffness variation of the square shaft rotor.

It can be seen that the horizontal stiffness of the square-axis rotor identified by using the method in this paper consists of two parts: constant value and periodic time-varying, and the identified stiffness curve has some burrs due to the presence of noise in the data acquisition. The frequency domain analysis of the curve is shown in Figure 18.

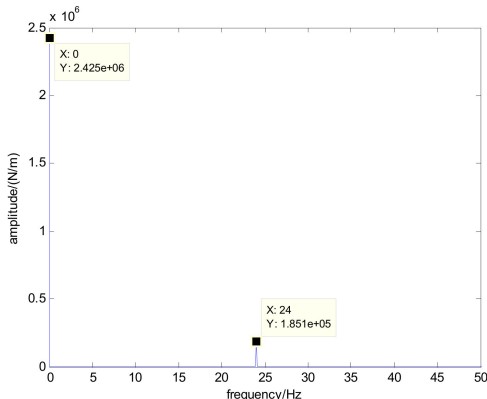

**Figure 18.** Square-axis rotor horizontal stiffness spectrum.

As can be seen from Figure 18, the stiffness identified in this paper is consistent with those values calculated as $k_c = 2.45 \times 10^6$ N/m and $k_v = 0.187 \times 10^6$ N/m, using the ANSYS finite element software. The variation of the horizontal stiffness of the square-axis rotor has obvious periodicity, and the frequency component $f = 24$ Hz is exactly two times that of the rotor rotation frequency $f_n = 720/60 = 12$, which is consistent with the time-varying quadratic stiffness of the rotor rotation.

## 5. Discussion

(1) The recursive formula for the identification of the periodic time-varying rotor system parameters derived in this paper has the advantages of concise structure, time-saving calculation, and high accuracy.

(2) The signal noise has a certain influence on the recognition result, so the signal-to-noise ratio should be improved as much as possible in the signal acquisition process. In the case of the Newmark-$\beta$ method used in solving the dynamic response of the system, the calculation step length affects the accuracy of the calculation. According to the sampling theorem, the sampling frequency should be greater than the analysis frequency of the signal by a factor of two or more, in order to avoid the occurrence of mixing. The smaller the calculation step, the better the recognition effect, but the corresponding calculation volume also increases, so an appropriate step size should be selected according to the actual requirements. At the same time, according to the results of Table 2 in line 230, it can be seen that when $h = 10^{-3}$ s, the accuracy is poor; when we take $h = 10^{-4}$ s, the accuracy has met the requirements, while if we take $h = 10^{-5}$ s, the computational volume is large; taking this into account, we chose $h = 10^{-4}$ s.

(3) By identifying the parameters of the numerical simulation model and the actual periodic time-varying rotor system, the time-varying stiffness parameters can be identified more accurately, thus verifying the correctness and effectiveness of the method. Compared with the state-space wavelet method and with the EMD method, the recognition accuracy of this paper is higher under the conditions of noise-free and high signal-to-noise ratios; the speed of identification by this method is improved by 53.38% compared to the state-space wavelet method, and the speed of identification by the EMD method is improved by 62.87%.

(4) This paper provides a new method for the parameter identification and fault diagnosis of engineering machinery structures with periodic time-varying parameters, such as gears and cracked rotors, which has great engineering application value.

**Author Contributions:** Conceptualization, Z.W.; methodology, J.T.; software, J.W.; validation, J.W.; formal analysis, J.W.; resources, Z.W.; data curation, J.W.; writing—original draft preparation, J.W.; writing—review and editing, Y.L.; visualization, J.W.; supervision, Y.L.; project administration, Z.W.; funding acquisition, Z.W. All authors have read and agreed to the published version of the manuscript.

**Funding:** This research was funded by the National Natural Science Foundation of China, grant number 52205115, the National Natural Science Foundation of China, grant number 12172231, and the Department of Education of Liaoning Province, grant number JYT2020158.

**Institutional Review Board Statement:** Not applicable.

**Informed Consent Statement:** Not applicable.

**Data Availability Statement:** Not applicable.

**Conflicts of Interest:** The authors declare no conflict of interest.

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
