# Peer review of "Parameter Identification Method for a Periodic Time-Varying System Using a Block-Pulse Function"

_aerospace, doi:10.3390/aerospace9100614_

Round 1
Reviewer 1 Report
-As authors mentioned in the abstract about the saving time of the proposed scheme, it seems a bit lacking support evidence. It would be nice if author can overcome it in the revision.
-Currently, they have some algorithm for parameters or functions estimations such as NN
| https://doi.org/10.1103/PhysRevD.103.036001 |
Data driven https://doi.org/10.1016/j.ins.2021.06.051,
Monte Carlo methods 10.1186/s13634-020-00675-6
and so on. Could authors discuss about them in order to emphasize the advantage of the proposed approach.!
- Furthermore, please verify some TeX errors such that page 3 line 108.
- Does it require the assumption for the compact set of the state x(t) \in \Omega_x in order to obtain the solution proposed by this work?
- The experimental results seem sufficient but the conclusion needs to be revised according to the results and the main contributions of this paper.
Round 2
Reviewer 1 Report
Authors already revised and considered for all reviewers' comments.